# The Second Side of the Coin—Resilience, Meaningfulness and Joyful Moments in Home Health Care Workers during the COVID-19 Pandemic

**DOI:** 10.3390/ijerph19073836

**Published:** 2022-03-23

**Authors:** Doris Gebhard, Julia Neumann, Magdalena Wimmer, Filip Mess

**Affiliations:** Department of Sport and Health Sciences, Technical University of Munich, Georg-Brauchle-Ring 62, 80992 Munich, Germany; j.neumann@tum.de (J.N.); magdalena.wimmer@tum.de (M.W.); filip.mess@tum.de (F.M.)

**Keywords:** home health care worker, health-related resources, resilience, meaning of work, joyful moments, Germany

## Abstract

Nursing literature predominantly focuses on job demands but is scarce for resources related to nurses’ work. In the face of the COVID-19 pandemic, resources that can buffer the health-impairing effects of increased demands gain importance. The aim of this study is to explore resilience, meaning of work and joyful moments in home health care workers in South Germany during the pandemic. Resilience and meaning of work were measured quantitatively; moments of joy were investigated qualitatively by audio diaries and analyzed with qualitative content analysis. In all, 115 home health care workers (mean age = 47.83 ± 11.72; 81.75% female) filled in the questionnaires and 237 diary entries were made by 23 persons (mean age = 46.70 ± 10.40; 91.30% female). The mean scores of resilience (5.52 ± 1.04; 1–7) and meaning of work (4.10 ± 0.92; 1–5) showed high levels, with significantly higher values in females. Home care workers experienced joyful moments 334 times in 60 different types in the categories of social relationships, work content, work organization, work environment and self-care. A deeper understanding of resilience, meaning of work and joyful moments provides a basis for the development of worksite health promotion programs that address both demands and resources in home health care workers.

## 1. Introduction

Examining literature on working conditions in nursing through the lens of salutogenesis may result in the estimation that researchers fail in “asking the right question” as Aaron Antonovsky proposed it [1]. For several decades, nursing literature has predominately focused on describing the prevalence of work strains and their consequences rather than exploring work-related resources [2]. Even though signs of a paradigm shift were observed during the last years, the long tradition of the biased view on demands still shapes the societal and self-perception of the nursing profession [3]. Moreover, it influences the scientific discourse within the field of worksite health promotion, indicated by a shortage of articles where intervention strategies aim at the promotion of attitudes related to the field of positive psychology [2]. Focusing on demands while marginalizing positive aspects of being a nurse contradicts the salutogenetic approach and thereby overlooks ways of improving nurses’ work life. To apply the salutogenetic principles, which strive to understand the underlying mechanism of (positive) health development at work [4], to nursing would broaden the leading pathogenic orientation. This extension of perspectives is theoretically based on the job demands–resources model (JD-R model) as the health-enhancing effects of job resources stand equally beside the adverse health effects of job demands [5,6,7].

### 1.1. Resources within the Job Demands–Resources Model

The range of investigated resources within the JD-R model is wide as the model has a broad scope, which assumes that any resource may affect employee health and well-being [8]. Initially, the JD-R model only considered work-related resources, defined within the model as physical, psychological, social or organizational characteristics of the job, that may be functioning in (1) achieving working goals, (2) reducing physiological and psychological costs of job demands and (3) stimulating personal growth and development [7]. Examples of work-related resources are team cohesion, performance feedback, autonomy and safety climate [8]. In revised versions of the model, personal resources are also integrated and defined as “psychological characteristics or aspects of the self that are generally associated with resilience and that refer to the ability to control and impact one’s environment successfully” ([8] p. 49); examples are hope, optimism, self-efficacy and resilience. 

Resources initiate two mechanisms within the JD-R model. Firstly, they trigger extrinsic and intrinsic motivational processes and may lead (e.g., via work engagement) to positive work-related outcomes and the promotion of health. Secondly, they buffer the impact of job demands on strain and thereby inhibit negative outcomes on organizational and individual levels. That means employees with a high degree of job resources may cope better with their job demands and have a better state of health [6]. This buffering mechanism may be particularly useful for occupational settings where reducing or redesigning demands is challenging, such as nursing [9]. 

Since the introduction of the JD-R model, its mechanisms were confirmed also within nursing-specific JD-R models [6,10,11]. Due to the high diversity of resources investigated, Broetje, Jenny and Bauer [12] synthesized nursing resources along the lines of the JD-R model. The authors identified six key job resources for nurses: (1) supervisor support, (2) fair and authentic management, (3) transformational leadership, (4) interpersonal relations, (5) autonomy and (6) professional resources. Personal resources are not represented within this list, suggesting they may have received limited attention in nursing research to date [12]. Their neglected role in current literature has also been shown in other reviews on resources in nursing, where personal resources were hardly featured [13,14]. Nevertheless, there is a growing interest in personal resources related to positive psychology [15] as they have shown to be effective in buffering work demands in nursing [16]. Two examples of personal resources, for which mechanisms of JD-R were confirmed in the nursing workforce, are resilience and meaning of work [17,18]. Moreover, an emerging field of research suggests that the experience of positive emotions at work are not just resources in themselves but also means to develop resilience and meaning within work [19].

### 1.2. Resilience and Meaning of Work

Resilience in nursing has become an important personal resource [18]. Resilience is a process of effectively negotiating, adapting to or managing significant sources of stress or trauma [20]. Within the nursing profession, it is defined as a process that “enables nurses to positively adapt to workplace stressors, avoid psychological harm and continue to provide safe, high-quality patient care” ([21] p. 15). Drawing from the JD-R model, current reviews confirm that nurse resilience mitigates job demands and their effects, including work stress, burnout, fatigue, anxiety and depression [18,22].

Meaning of work can be defined as “the discovery of existential meaning from work experience, work itself and work purpose/goals” ([23] p. 2259), and it is related to four critical attributes: (1) experiencing positive emotion at work, (2) meaning from work itself, (3) meaningful purpose and goals of work and (4) work as a part of life towards meaningful existence [23]. A qualitative study [24] identified four main topics that bring meaning into the work of the nursing profession: (1) relationships with colleagues and patients, (2) compassionate caring, (3) sense of identity through the profession and (4) mentoring culture. Meaning of work was found to mediate the effects of stress and emotional exhaustion in health care professionals [17,25].

### 1.3. Joyful Moments

The experience of positive emotions is not only an attribute for the meaning of work; it is also associated with the development of resilience. The broaden-and-build theory [19] posits that positive emotions broaden the thought–action repertoires of an individual in the short term, which in turn contributes to building enduring personal resources. Thus, positive emotions can initiate upward spirals toward enhanced emotional well-being by triggering momentary broad-minded coping mechanisms which are associated with the building of resilience [19,26]. Within the spectrum of positive emotions, joy has a special significance in health care. Gergen Barnett [27] described that the joy of practice of those who deliver care is critical to the future of health care; it is at the center of health care delivery “from which all other aims of health care must stem” ([27] p. 1032). To determine the meaning of joy in the discipline of nursing, Cottrell [3] conducted a concept analysis. The author concluded that joy can be described as a positive concept that consists of an intense, sudden and transient loss of self, followed by a deep connection (to nature, god and/or others), which is involuntary, provides insight and has associated physical characteristics. Therefore, the concept of joy goes beyond hedonic emotions, has a self-transcending character and allows individuals to flourish. In interviews with nurses, Wilkes et al. [28] identified relationships with others and the working system as the two major factors affecting joy in nursing. Within the factor relationships, the ability to do and share with others, connecting with others, supporting others and educating others were mentioned; regarding the working system, having a broad scope of practice, management support and opportunities to learn were mentioned as factors affecting joy positively. Based on these findings, the same authors identified determinants of enjoyment in hospital nursing [28]. Most positively associated with joy at work were the aspects educating others, connecting with others, variety of work, doing and sharing with others and supporting others. So far, the experience of joyful moments in nursing and its determinants have not been explored well, and research is limited to institutional settings of care [29]. 

### 1.4. Resources in Times of COVID-19

An increase in work-related demands during the COVID-19 pandemic is undisputed and well documented, and it is leading to a deterioration of health in the nursing workforce [30,31,32]. With the challenges of the pandemic and its severe impact on nursing working conditions, a renewed strengthening of the pathogenic approach can be observed, as nurses’ resources are described less frequently [33]. A considerable number of empirical research studies related to nurses’ mental health status during the COVID-19 pandemic have been published [30,31,32]. The majority of the studies set the focus on the negative effects of the pandemic-related demands on mental health outcomes such as depression, anxiety, burnout and insomnia, and there is a lack of reporting on resources and their health-enhancing effects in the pandemic situation [30,31,32]. Nevertheless, under these circumstances, the strengthening of personal resources in nurses, in line with the mechanisms of the JD-R model, became more essential than ever before [34]. 

In the face of a pandemic, the concept of resilience especially gained attention in nursing research and was identified as an effective antidote against pandemic-related demands [35,36,37]. Besides the research on resilience, literature on other personal resources in nursing during the COVID-19 pandemic is scarce. To the authors’ knowledge, the two studies from Blanco-Donoso et al. [38] and Mojtahedzadeh et al. [33] are unique in exploring resources in nurses working in long-term care during the COVID-19 pandemic with a qualitative approach. Blanco-Donoso et al. [38] found high levels of professional satisfaction; identified social support as the most important resource; and documented nursing home workers’ positive feelings of gratitude, professional fulfillment and sense of pride as work-related positive psychological consequences of the pandemic. In their interview study, Mojtahedzadeh et al. [33] showed that colleagues have a central role as work-related resources within the four emerging categories: (1) perceived social support and the feeling of trusting; (2) intensified communication with colleagues and superiors with special regard to (3) exchange of information; and (4) an increased appreciation, first and foremost from colleagues and superiors, but also from patients and neighborhoods. These two studies support the assumption that even in the presence of a pandemic, nurses experience resources, and the results give a first insight into their concrete occurrence. Beyond the pioneer character of Mojtahedzadeh et al.’s [33] work by reporting besides demands also resources in the pandemic situation, the care setting they explored is a unique characteristic. So far, home health care workers are neglected in nursing literature (before and during the pandemic), although their importance is rapidly raising within the health care sector. 

### 1.5. Home Health Care

Home health care services are the fastest growing branch in the long-term care sector in Germany. From 2017 to 2019, the formal home health care sector increased by 18%, whereas the institutional long-term care sector (e.g., nursing homes) remained stable within the same period [39,40]. Similarly, in other Western countries, the general trend clearly indicates a future growth of this branch, as the U.S. Bureau of Labor Statistics predicts an increase in the need for home health care workers by up to 33% in 2030 [41]. Reasons for this development can be found in the general increase in people in need of care combined with the preference to remain in the familiar environment of their own home, as well as policy funding priorities [42]. Additionally, home health care services have become an increasingly attractive alternative for patients since the COVID-19 pandemic has overwhelmed hospital systems and nursing homes [43]. 

While home health care workers perform similar care tasks as their colleagues in the institutional long-term care sector, the contextual factors differ. Working alone, mobile and in the home environment of the person in need of care entails, for example, a high degree of individual responsibility and autonomy, indirect shift handover and the need for route planning [44,45]. Compared to other nursing settings, home health care workers seem to face the most challenging working conditions [46]. Nevertheless, a focus group study identified three specific resources which make it attractive to work in home health care [47]: (1) being a “linchpin”, in the sense of having a central position in the community and within the care of the patient; (2) autonomy, over patient care, in their own work and within work organization; and (3) variety, in patient situations and working tasks. Furthermore, Kusmaul et al. [42] showed in an interview study that home health care workers perceive a high degree of empowerment within their work. The feelings of competence, autonomy, having a positive impact on patients’ lives and performing meaningful work are aspects reported in relation to psychological empowerment. Those first insights into resources of home health care workers are rare literature highlights as current evidence is scarce and further research is urgently needed [33]. 

### 1.6. Study Aims

Against this background, the aim of this study can be derived from the following research gaps: (1) There is a predominance in nursing literature on job demands, while literature investigating resources (work-related and personal) related to nurses’ work is insufficient; the COVID-19 pandemic reinforces this biased view. (2) Research on personal resources in nursing, in particular, is still scarce, even though their health-enhancing and protective effects were verified within the JD-R model. (3) Resilience, meaning of work and joyful moments show high potential within the mechanisms of the JD-R model but are insufficiently investigated in nursing. (4) All these shortcomings are more obvious when focusing on an increasingly critical setting of health care, the home health care sector.

Consequently, the three aims of this study investigating home health care involve (1) the state of resilience and meaning of work; (2) the types and frequencies of joyful moments; and (3) the differences in resilience, meaning of work and joyful moments with regard to gender, age, native language, region and profession. 

## 2. Materials and Methods

A convergent mixed-methods research design was applied. Qualitative and quantitative data were collected and analyzed separately during a similar timeframe [48]. Data integration was performed through merging [49]. The mixed-methods findings are represented using joint displays. Appendix A presents the mixed-methods research design within a procedural model [50].

The study is part of the project “EMMA” (https://www.sg.tum.de/en/sportdidaktik/forschung/emma/; accessed on 19 March 2022). Six home health care services participate in the project. Three of them are located in the city of Munich (Germany); three are located in rural areas of South Germany. All employees (n = 200) were invited to participate in the quantitative survey. For the qualitative survey, the managers were asked to choose four employees who were willing to participate. Thus, a convenience sample of 24 employees of home health care services was recruited.

### 2.1. Quantitative Methods

#### 2.1.1. Data Collection

In September and October 2020, an information event was implemented, to introduce the participating home health care services to the project’s aims and process. At the end of the event, additional time was provided to fill in the questionnaires. For those employees who were not able to participate in the information event, blank questionnaires and a questionnaire box were placed in all services for two weeks. 

#### 2.1.2. Quantitative Measures

Resilience was measured using the short version of the German Occupational Resilience Scale [51]. It includes eight items rated on a 7-point Likert scale (1 = not true, 7 = true) across four sub-scales: (1) emotional coping, (2) comprehensive planning, (3) positive reframing and (4) focused action. Higher values indicate a higher degree of resilience. 

Meaning of work was measured using the German version of the Occupational Meaningfulness Scale [52,53]. It includes six items rated on a 5-point Likert scale (1 = not true, 5 = true). Higher values indicate a higher degree of meaning of work. 

Additionally, the questionnaire included questions on sociodemographic data regarding gender, age, region, native language and profession. 

#### 2.1.3. Quantitative Statistical Analysis of Quantitative and Qualitative Data

Differences in resilience, meaning of work and joyful moments were tested regarding gender, age, region, native language and profession. Qualitative data were converted into a dichotomous variable, indicating if a diary entry includes a mention related to the category or not. A Shapiro–Wilk test was conducted to test the distribution of variables. As all variables were non-normally distributed, a Mann–Whitney U test or a Kruskal–Wallis H test was applied. A chi-square test was conducted for nominally scaled variables.

The differences were considered statistically significant with *p* < 0.05 for all analyses. For statistically significant results, the effect sizes were calculated. The Pearson correlation coefficient (*r*) was calculated if the Mann–Whitney U test or the Kruskal–Wallis H test were conducted; phi (*w*) was calculated if a chi-square test was conducted. According to Cohen (1988) [54] the effect sizes were interpreted as follows: 0.1 ≤ |*r*| or *w* < 0.3 = small; 0.3 ≤ |*r*| or *w* < 0.5 = medium; |*r*| or *w* > 0.5 = large. Statistical analysis was performed using SPSS Statistics (IBM Corporation, Version 23.0. Armonk, NY, USA).

### 2.2. Qualitative Methods

To investigate joyful moments in home health care workers´ daily working routine, the audio diary method was applied. The use of audio diaries provides an opportunity to capture situated events and feelings as they occur in a daily work routine, independent of place and time. Unlike conventional diary methods, diary entries are not written but are verbal. This method has been used to explore various aspects of the lived experience of health care professionals [55,56]. From the perspective of work psychology, audio diaries are suggested to be used especially for the examination of fluid and transient working patterns of underexplored employee populations [57], which applies to the target group of home health care workers, especially under COVID-19-related working conditions.

#### 2.2.1. Data Collection

Participants were instructed to make diary entries directly after 12 consecutive shifts. They received an audio diary package with the following content: (1) a digital voice recorder, (2) a simple step-by-step guide (pictures and text) on how to use the digital voice recorder, (3) a prompt sheet on what to record in the diary and (4) a short questionnaire on sociodemographic variables (gender, age, native language, region and profession). The instructions about the diary entries were outlined as follows: *Please think about your workday. Which moments were joyful and why?*


#### 2.2.2. Data Analysis

Transcripts were analyzed using qualitative content analysis based on Mayring [58]. Deductive as well as inductive procedures were used for the development of the categories. Firstly, the technique of content-related structuring was used through the application of five predefined categories: (1) social relationships, (2) work content, (3) work organization, (4) work environment and (5) new forms of work. Due to the absence of defined areas of resources in home health care workers, these categories were taken over from the field of work-related psychological strains in this target group [45]. Secondly, the technique of summarizing was used to define sub-categories inductively. The category system and the coding agenda were steadily revised and complemented. In addition, two coders conducted a reliability proof of the categories based on the entire material. Disagreement was dissolved by discussion and consensus. MAXQDA 2020 (VERBI Software, Berlin, Germany, 2019) [59] was used for qualitative data analysis. 

## 3. Results

### 3.1. Sample

In all, 115 home health care workers filled in the questionnaire (57.5% response rate), and 23 participants produced 237 diary entries. The audio diary sample was partly represented within the questionnaire sample, as 14 home health care workers participated in both surveys. Table 1 shows the sample characteristics. 

### 3.2. State of Resilience and Meaning of Work

Table 2 presents the results for meaning of work and resilience. Quotes from audio diaries give an insight into the lived experience of joyful moments related to resilience and meaning of work.

### 3.3. Types and Frequencies of Joyful Moments

Within 33 out of 237 diary entries, participants reported the absence of joyful moments. Therefore, the results are based on 204 diary entries. Home health care workers identified 60 different types of joyful moments. In sum, these joyful moments were mentioned 334 times. Figure 1 presents the distribution and frequencies of joyful moments in the applied deductive categories (1) social relationships, (2) work content, (3) work organization and (4) work environment. No mention was found within the deductively applied category new forms of work; the category self-care was developed inductively based on the material.

The category social relationships includes 31 inductively developed sub-categories. Within these categories, joyful moments related to patients were mentioned most often, followed by joyful moments referring to the team and relatives of the persons in need of care. The two categories positive interaction and appreciation were identified in relation to all three groups of persons, with a predominance of frequencies in the category patients. Support was mentioned as joyful with regard to colleagues and relatives; gratitude was mentioned as joyful with regard to patients and relatives. Figure 2 shows the categories’ structure up to level three (level four is presented in Appendix B) and the number of mentions for each category. Quotes give an insight into the participants’ narratives.

Within the deductive category work content, nine sub-categories were inductively developed. Joyful moments related to a sense of achievement were mentioned most often. These joyful moments referred to the successful mastery of challenging situations, classified within quantitative challenges and qualitative challenges. Having an easy working day without any complications and valuing one’s own work were mentioned second and third most frequently, respectively. Furthermore, subjects reported experiencing varied, interesting and new work contents as joyful. All categories up to level three (level four is presented in Appendix B), frequencies of mentions and related quotes are presented in Figure 3.

The category work organization comprises 11 sub-categories. Aspects related to an easy schedule, such as no time pressure, earlier or timely end of work or later beginning of the working day, were reported most often. These aspects were followed by joyful moments referring to the feeling of being pleased with the planning of the roster and the route, having appropriate working materials and there being more staff on duty than usual. Figure 4 illustrates the categories’ structure, the number of mentions for each category, and related statements.

The category work environment includes five inductively developed sub-categories, all related to being on the road. Within these categories, joyful moments related to a quiet traffic situation were mentioned most often, followed by good weather and a well-prepared car. The inductive category self-care includes joyful moments related to an active lunch break, self-encouragement, feeling fit and having a joyful anticipation of one’s evening. 

The category structure, frequencies of mentions and related quotes of work environment and self-care are presented together in Figure 5, as the category structures include only two levels.

### 3.4. Differences in Resilience, Meaning of Work and Joyful Moments

Table 3 presents the differences in resilience, meaning of work and joyful moments related to gender, age, native language, region and profession. Mean values and standard deviations are presented for the questionnaire data. Percentages of diary entries that include a quote related to the category are presented for the audio diary data (e.g., in 75.5% of diary entries produced by women, joyful moments related to the category relationships were mentioned). Effect sizes are presented for statistically significant differences. Statistical coefficients are presented in Table A1 (Appendix C). Significant differences related to gender were obtained for resilience (small effect size), meaning of work (medium effect size) and work content (small effect size), each with higher values for female participants. For age, significant differences were found in work content (small effect size), with percentages decreasing with increasing age. Participants working in rural regions showed significantly higher values in meaning of work and mentioned joyful moments related to relationships significantly more often (small effect size), whereas work content was mentioned significantly more often in diary entries recorded by employees in urban regions (small effect size). Between health care workers and administrative staff, significant differences were obtained for the mention of joyful moments related to relationships (small effect size), with higher values for health care workers, and related to self-care, with higher percentages of self-care-related diary entries in administrative staff (medium effect size). No differences were found for native language.

## 4. Discussion

This study contributes to the understanding of resilience, meaning of work and joyful moments in home health care during the COVID-19 pandemic. 

The mean scores for resilience and meaning of work were significantly higher in female home health care workers than in males. In addition, participants working in rural areas showed significantly higher levels in meaning of work than their urban colleagues did. 

Home health care workers experienced joyful moments 334 times in 60 different types in the categories of social relationships, work content, work organization, work environment and self-care. Joyful moments related to social relationships were mentioned most often, predominantly in interaction with patients. Being a care worker and working in a rural area was associated more frequently with experiencing joy related to social relationships. Experiencing joy regarding the work content most often occurred due to a sense of achievement. Home health care workers who are female, younger and work in an urban region were more likely to experience joyful moments related to work content. Within the context of work organization, an easy schedule was primarily mentioned as a reason for joy. Related to the work environment, a quiet traffic situation was the aspect mostly associated with joy. To experience joyful moments in the context of self-care was an aspect newly identified in this study. Those moments occurred typically during an active lunch break and were found to be primarily experienced by the administrative staff.

### 4.1. Resilience and Meaning of Work

The mean score of resilience found in the present study was higher than that in a sample of German employees of different professions (5.52 vs. 5.06) [60]. Comparing all mean scores of the sub-scales, home health care workers again showed consistently higher values than students [51] and a sample of employees mostly working in the private sector [61]. Contrary to previous findings in nursing [62,63], no differences were found between the age groups and the levels of resilience. Previous studies showed inconsistencies regarding differences between female and male nurses. Consistent with this study, female nurses showed significantly higher levels of resilience in Ren et al.’s trial [64], whereas Zheng et al. [63] showed marginally higher values in males. Ang et al. [62] found no gender differences in resilience. 

Present results show higher mean scores for the sub-scales comprehensive planning and focused action than for the sub-scales emotional coping and positive reframing. Comprehensive planning and focused action refer to a problem-focused coping strategy, in contrast to the other two sub-scales which represent emotion-focused coping strategies [51]. This leads to the assumption that home health care workers have higher competencies in tackling work-related challenges through proactive planning, focusing on the problem, searching for different solutions and pursuing them persistently when problems occur [51]. This explanation is supported by the results from the audio diaries. More than half (30 mentions) of joyful moments related to the working content referred to a sense of achievement when mastering challenging situations. The high frequency of mentions indicates that finding ways to deal with challenges is part of the daily working routine in home health care work. An observational study on home health care workers [65] showed that nurses develop different strategies for organizational challenges and lack of time. Strandas et al. [63] observed that nurses frequently juggle minutes between patients and manipulate the timer that documents the time needed for every visit, as nurses have an exact amount of time that is assigned to every single patient. The observed nurses stated that their focus is on patients and not on the fulfillment of predefined tasks and budgets. Taking these findings together with the results of the present study, resilience in home health care may be discussed in terms of being able to provide good care despite the organizational system, and not because of it. However, experiencing joy when overcoming challenges successfully is a new but frequently found aspect that has not yet been identified by other studies on joy in nursing [28,29,66,67].

This approach fits perfectly with the image of “the hero in the failing system”, as such home health care workers were characterized in a focus-group study from Timonen and Lolich [68]. Focus-group participants stated that only a hero or a saint would take this type of job. Home care workers themselves described the fundamental drive behind their efforts not as doing something heroic but as a sense of joy and meaning in doing a good job [65]. Audio diary entries in this study confirmed this statement, as the feeling of doing something meaningful, fulfilling and good was mentioned as a source of intrinsic reward that can balance the high efforts of care work [69]. As studies indicated that nurses perceive an imbalance between efforts and rewards [70,71], diary entries showed that high meaningfulness is set in relation to the low payment, indicating a compensating function of intrinsic rewards when extrinsic rewards are lacking. A representative German study compared care workers and employees in all other professions in terms of fair payment and meaning of work [72]. More than three-quarters of the geriatric care workers (78%) had the feeling of unfair payment, whereas not even half (48%) of employees of all other professions shared this opinion. In contrast, the vast majority (94%) of geriatric care workers believed that they make a meaningful contribution to society, but only two-thirds (67%) of employees in all other professions came to the same conclusion [72]. 

However, the quantitative results on the meaning of work in the present study only partially support these findings. Compared to other occupational settings, the mean score of the meaning of work is higher than that for the employees in the insurance industry (4.10 vs. 3.61) [52] but lower than that for the employees in non-profit organizations (4.44) [73]. Compared to the residential geriatric long-term care sector in Germany, home health care workers in this study showed a lower mean score in the meaning of work than their colleagues working in nursing homes did (4.42) [53]. The reasons for the comparatively lower level of the meaning of work can be found in the specific working conditions in home health care, which may function as barriers for meaning of work. Weckmüller et al. [53] discuss provisions instead of autonomy, lack of appreciation and bureaucracy as the most relevant barriers for the meaning of work in nursing. In addition, Hognestad Haaland et al. [74] found that ethical dilemmas at work are negatively related to the meaning of work in nursing. Barriers related to provisions and ethical dilemmas may be particularly true for home health care. Home care workers have to work within a tighter schedule than nurses working in residential care [46]. Frequently, they have to visit five patients within 20 min, including time for transportation and documentation [65]. Several home health care workers feel bound by the strict time frame [65], which limits the autonomy of working alone. A systematic review [75] summarized ethical dilemmas in nursing in the following three themes: (1) balancing harm and care, (2) work overload affecting quality and (3) navigating in disagreement. While all of these themes occur in home health care, there is a significant difference: many patients live alone, and the home health care worker is sometimes the only person they see all day. Therefore, the home health care workers also feel responsible for the needs of the patients that lie outside the allocated time and paid tasks. As no time is allocated for emotional support [68], they initiate conversations while performing care [65]. Moreover, patients frequently ask to perform additional tasks outside of the job. Knowing that no one else will do the task for the patients makes it difficult for home health care workers to say “that’s not my job” [76]. These few insights into the practice clearly show that even though the meaning of work is an immanent value in home health care, barriers may inhibit home health care workers from experiencing it. 

### 4.2. Joyful Moments

Audio diary results indicate that social relationships are the most important source of joy in home health care. This result is partly in accordance with the existing research on joyful moments in nursing, as Wilkes et al. [65] identified relationships with others as one of two major factors affecting joy in nursing [27]. In contrast to Wilkes et al.’s findings [65], the present study also identified the patients’ relatives as a relevant person group that is involved in the occurrence of joy. This aspect may be unique for the home health care setting, as a good collaboration between the home health care worker and the family caregiver is crucial for good care [77]. The current results indicate that a reciprocal relationship between home health care workers and relatives is not just a family carers’ need [77]; it is also positively experienced by the care worker and, moreover, a source of joy. 

Unfortunately, Wilkes et al. [67] do not present a differentiated analysis of their results or a detailed explanation of identified topics beyond using quotes. However, the topics “to do and share with others”, “connect with others” and “supporting others” identified by Wilkes et al. [67] may be comparable to the category of social relationships with patients in this study. Within this category, positive interaction with patients was the sub-category mentioned most often within the audio diaries. Having fun together was the most prominent topic related to positive interactions with patients, but it was also mentioned frequently with regard to the home health care team and relatives (see also Appendix B). Humor as an expression of positive emotions between patients and home health care workers was accordingly observed by Heyn et al. [78]. Moreover, humor was found to be a useful strategy for improving social relations or dealing with stressful situations in nursing, as Navarro-Carillo et al. [79] explored: positive humor styles showed positive associations with happiness, sociability, hope and life satisfaction in nurses. 

The second most mentioned aspect of joy within the category positive interaction with patients was understanding/empathic patients. Consistent with these results, Strandas et al. [65] also observed empathic behavior of home health care patients. Patients tended to lower their demands and adjusted their behavior with the intention to ease the burden on home health care workers. The current results indicate that home health care workers appreciate this behavior, which can be critically discussed. 

Study results clearly indicate that joy in home health care workers is frequently associated with the positive emotions of patients. The observational study from Heyn et al. [78] showed that the expression of positive emotions within home care visits occurs reciprocally and frequently between nurses and patients, giving a mean of 19 expressions per visit. 

Appreciation and gratitude shown first and foremost by patients, but also by relatives and the team, was found to be a frequently mentioned reason for joyful moments in this study. Thus, diary entries indicated again the high value of intrinsic rewards when extrinsic rewards are lacking. According to the current findings, Mojtahedzadeh et al. [33] identified appreciation from colleagues, supervisors and patients as an important resource for home health care workers. 

Looking closely at the situations in which home health care workers experienced moments of joy in their relationships with patients indicates that most carers and patients become emotionally involved. This assumption is in accordance with the observations of patient–home carer relationships made by Strandas et al. [65]. However, the discussion about the type of relationship the carer and the patient should have in home care is characterized by ambivalence [68]. At first sight, a paid care worker is neither family nor a friend, but in many cases, close long-term relationships evolve. Persson et al. found that a positive and close relationship in home care may facilitate a mutual understanding of each others’ situations and is associated with home care workers’ health [80].

Besides social relationships with patients, the current study shows the strong impact of the team on home health care workers’ joy. This generally confirms previous results on joyful moments and resources [33,67]. In contrast to the results from Wilkes et al. [67] and Galuska et al. [66], which identified educating others as the most central source of joy related to the team, receiving support was found in the present study as the aspect most frequently related with joy. Thereby, the results clearly showed the special role of nursing trainees, as home health care workers often reported joy when they experienced the luxury of working in pairs. Furthermore, the predominance of the home health care team as the most important aspect regarding resources in home care [33] was not confirmed in the present study, as patients were mentioned more than twice as frequently. 

In this study, joyful moments related to work content were found to show the strongest relation to the topics resilience and meaning of work. Similar to the findings of Wilkes et al. [65], varied and interesting tasks and the opportunity to learn new things were mentioned as joyful within the category work content. Experiencing joy when having easy work content, e.g., when no complications occur during the shift, indicates that simply the absence of demands triggers the experience of joy among home health care workers. This assumption is further confirmed when exploring the results of joyful moments related to work organization. The majority of joyful moments occurred in relation to a normal working day without any special incidents, or in terms of a reduced workload. The results indicate that a “normal” workday is rare in home health care. This assumption is supported by the findings of Fjortoft et al. [81], who showed that it is normal for home health care workers to have unpredictable workdays. They face a high degree of urgency and time pressure as they must always be flexible when a number of different things are happening [81]. However, the unpredictability had also been found as something home health care workers like about their work [81]. 

The results do not show any joyful moment associated with the primary work environment in home care—the patient’s home. All aspects within the category work environment were associated with being on the road. Again, most joyful moments occurred because of the absence of well-known demands associated with mobile work, such as traffic jams or problems finding a parking space. Additionally, some new and positive aspects were found to trigger joy when being on the road. Good weather and a beautiful landscape make the way to patients joyful. These findings contribute to a more holistic picture of mobile work in nursing, as joy also occurs on the road. 

No diary entry was found to fit in the initially deductively developed category “new forms of work”, which was taken over from the field of work-related psychological strains in this target group [45]. This category was replaced by the inductively developed category “self-care”. Having an active lunch break was the joyful moment mentioned most often related to self-care. Again, this result indicates an absence of active lunch breaks on normal workdays in home health care. According to recent results for home health care workers in Germany, the majority reported a complete absence of breaks [33,82]. 

Even in times of COVID-19, all participants experienced joyful moments frequently, most of them on a daily basis. However, the current study did not find any mentions in the audio diary entries related to the COVID-19 pandemic. In contrast to the studies on resources in nursing during the COVID-19 pandemic, no work-related psychological consequences of the pandemic were reported as joyful, and also an increase in appreciation was not observed. Nevertheless, this does not indicate that the pandemic situation had no impact on the results. Markkanen et al. [83] found that hygiene regulations, such as physical distancing and masking requirements, undermine the nature of the patient–caregiver relationship as communication is made more difficult and less compassionate. Furthermore, Sun et al. [84] reported “growth under pressure” as a psychological phenomenon in their study on hospital nurses during COVID-19. The authors presented an increased affection and gratefulness, development of professional responsibility, and self-reflection within the interviewed nurses. These are two aspects that may also have affected the results of the current study.

This study shows that joyful moments are already present in the daily working routine of home health care workers, even though those moments may potentially go unnoticed within high workloads. From the view of worksite health promotion, this indicates the need for interventions aiming at facilitating the awareness of joyful moments rather than solely enhancing the occurrence of them. This approach implies a conscious shift towards already existing joy and refers to the ability to cultivate positive cognitions and emotions to gain momentum. The intervention “Three Good Things” is an example of the successful implementation of this approach among health care workers [85,86]. The concept is closely related to the audio diaries applied in this study, as participants had to keep a daily log of things that went well. This concept appears to be well suited for use in home health care, as it requires a minimal time commitment; is applicable without any additional skills training; and shows significant long-term improvements in exhaustion, depression, happiness and work–life balance [86]. 

### 4.3. Strengths and Limitations

This is the first study that investigates resilience, meaning of work and joyful moments in home health care workers. Considering that mixed-methods designs are not well established in nursing literature [87,88], the detailed presentation of the applied mixed-methods design may give a best-practice example that could be evolved through further research.

The main limitations of the study are related to the sample and the comparability of the results. The ability to generalize the results is reduced by the sample size and the convenience sampling which may have biased the results. Various types of resilience and meaning of work scales, as well as different versions of the applied scales, limit the comparability of the results. In addition, the lack of studies on joy in nursing and the missing details in the existing studies hamper a deeper discussion and the merging with existing evidence. 

## 5. Conclusions

It is crucial to emphasize that focusing on resources in home health care is not meant to be a way of avoiding discussions about demands, but rather an opportunity to see also the other side of the coin. As the results clearly showed, resources in home health care cannot be discussed without their relation to demands, as joy often occurs because of the absence of demands shaping home care workers’ work life. However, a deeper understanding of resilience, meaning of work and joyful moments provides a basis for the development of worksite health promotion programs addressing both demands and resources in home health care workers and aimed at facilitating the experience of joy in nursing practice. Finally, this study contributes to a more realistic image of home health care workers in society, as it shows that they are neither “the poor carer” nor “the hero in the failing system”.

## Figures and Tables

**Figure 1 ijerph-19-03836-f001:**
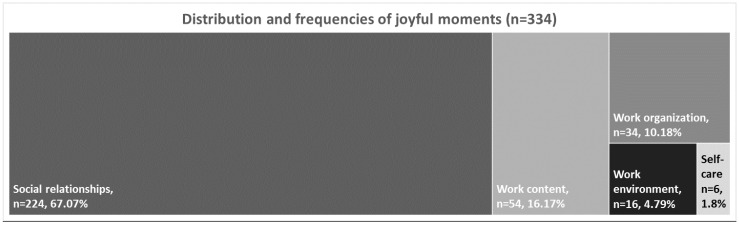
Distribution and frequencies of joyful moments.

**Figure 2 ijerph-19-03836-f002:**
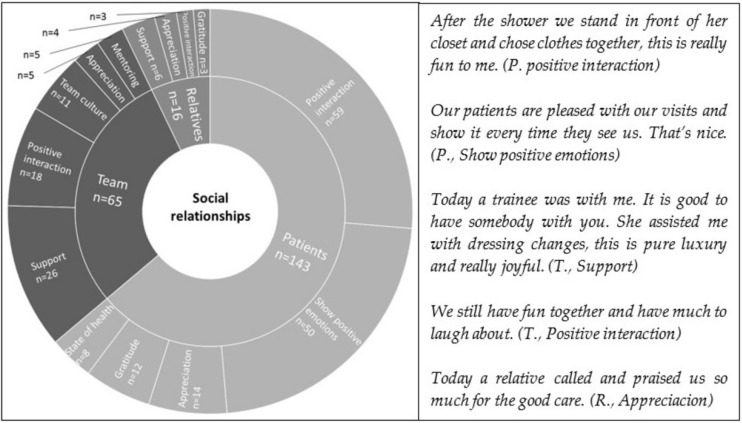
Distribution and frequencies of social relationships.

**Figure 3 ijerph-19-03836-f003:**
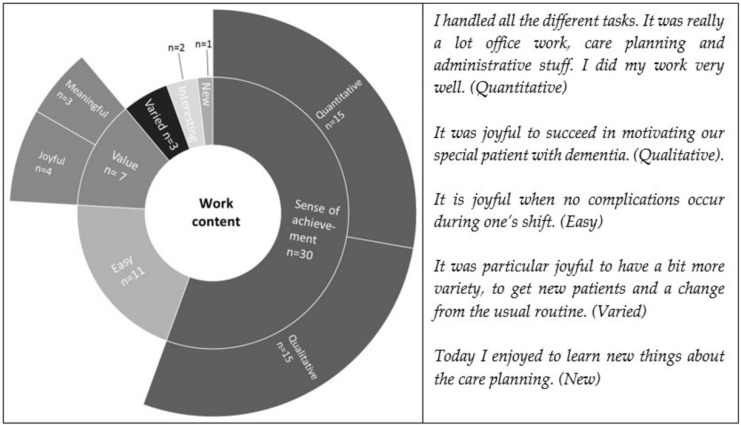
Distribution and frequencies of work content.

**Figure 4 ijerph-19-03836-f004:**
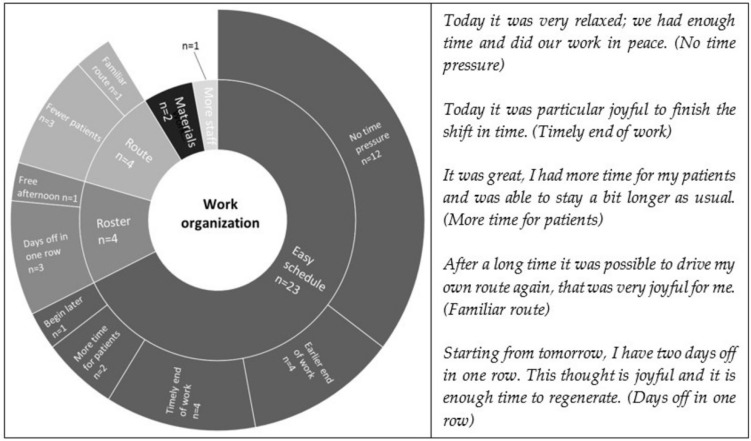
Distribution and frequencies of work organization.

**Figure 5 ijerph-19-03836-f005:**
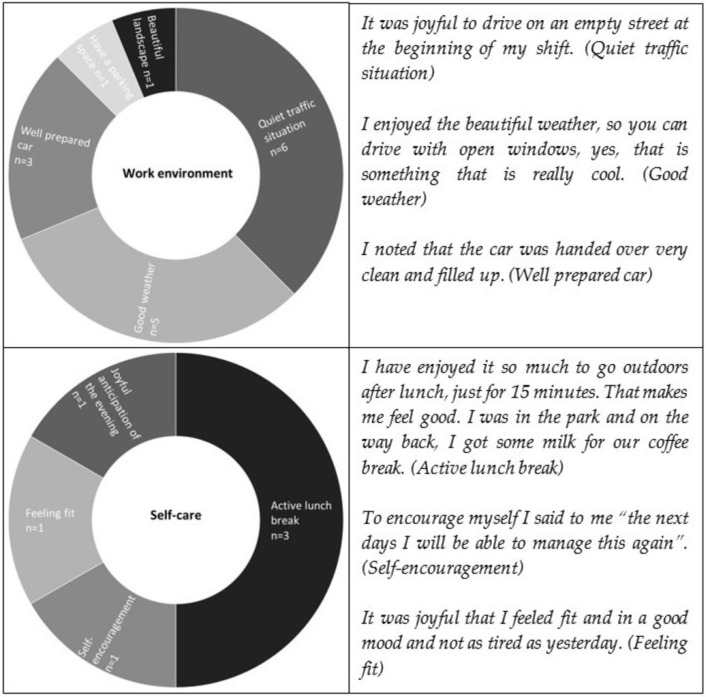
Distribution and frequencies of work environment and self-care.

**Table 1 ijerph-19-03836-t001:** Sample characteristics.

Variable	Questionnaire (n = 115)	Audio Diary, Participants (n = 23)	Audio Diary, Entries (n = 237)
Age, years, mean ± SD	47.83 ± 11.72	46.70 ± 10.40	46.22 ± 10.46
Gender, female (%)	81.74	91.30	91.50
Native language, German (%)	73.04	78.26	76.70
Region, urban (%)	46.96	47.83	49.20
Profession, care worker (%)	87.83	86.96	89.40

**Table 2 ijerph-19-03836-t002:** Joint display of resilience and meaning of work.

Variable	Mean ± SD (n)	Quotes
Resilience (OR) OR—Emotional Coping OR—Comprehensive Planning OR—Positive Reframing OR—Focused Action	5.52 ± 1.04 (n = 107) 5.46 ± 1.34 (n = 111) 5.96 ± 1.18 (n = 112) 5.17 ± 1.58 (n = 110) 5.55 ± 1.29 (n = 109)	*I was able to make my tour just in time, although it was challenging. That was joyful today.* *We have one special patient. He does not want to take a shower more often than every two or three weeks. Even then, it is difficult and he allows not all of us to help him. Today I managed that he takes a shower. Every time when I showered him successfully, I am proud. That motivates me.* *I just managed to complete my tasks, although many things happened unexpectedly.*
Meaning of work (OM)	4.10 ± 0.92 (n = 111)	*Today, all clients were very kind and they expressed their appreciation for the help we deliver. It is a really great feeling when you recognize that you make wonderful and right things.* *The situation is this, you make, and you are allowed to make, a very meaningful job, which is very fulfilling. A lot of money does not characterize it but meaningfulness. At the end of the day, you go home with a good conscience.* *You go home with the certainty, with the benevolent certainty, that today you did something good. You give lonely people who stay socially isolated at home a motivating reason to participate in society and help them to live their life in dignity.*

**Table 3 ijerph-19-03836-t003:** Differences in resilience, meaning of work and joyful moments.

Variable Heading	Quantitative				Qualitative									
	Meaning of Work		Resilience		Social Relationships		Work Content		Work Organization		Work Environment		Self-Care	
	Mean (SD)	*p*-Value	Mean (SD)	*p*-Value	%	*p*-Value	%	*p*-Value	%	*p*-Value	%	*p*-Value	%	*p*-Value
Gender														
Female	4.26 (0.79)	<0.001 *r* = 0.34	5.67 (0.92)	0.012 *r* = 0.24	75.5	0.496	20.1	0.028 *w* = 0.15	15.2	0.583	6.5	0.142	3.3	0.424
Male	3.3 (1.13)		4.94 (1.13)		68.4		42.1		10.5		15.8		0.0	
Age														
20–36	3.77 (0.86)	0.080	5.28 (1.24)	0.565	78.3	0.834	41.3	<0.001 *r* = 0.26	17.4	0.507	2.2	0.104	0.0	0.482
37–53	4.15 (0.98)		5.58 (1.03)		72.5		21.3		15.0		7.5		7.5	
54–70	4.19 (0.89)		5.55 (0.98)		75.3		11.7		13.0		10.4		0.0	
Native language														
German	4.11 (0.93)	0.877	5.63 (0.91)	0.144	76.4	0.345	20.4	0.258	12.7	0.130	7.6	0.798	3.8	0.178
Not German	4.08 (0.90)		5.21 (1.30)		69.6		28.3		21.7		6.5		0.0	
Region														
Rural	4.32 (0.74)	0.015 *r* = 0.23	5.70 (0.89)	0.089	83.2	0.007 *w* = 0.19	12.9	0.002 *w* = 0.22	13.9	0.714	5.9	0.432	1.0	0.100
Urban	3.83 (1.05)		5.31 (1.18)		66.7		31.4		15.7		8.8		4.9	
Profession														
Care worker	4.17 (0.85)	0.101	5.53 (1.06)	0.722	78.3	0.001 *w* = 0.22	21.7	0.631	15.6	0.383	8.3	0.150	1.1	<0.001 *w* = 0.30
Administrative staff	3.60 (1.23)		5.43 (1.02)		47.8		26.1		8.7		0.0		17.4	

## Data Availability

The data presented in this study are available on request from the corresponding author. The data are not publicly available due to restrictions of privacy.

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
