# Peer review of "The Second Side of the Coin—Resilience, Meaningfulness and Joyful Moments in Home Health Care Workers during the COVID-19 Pandemic"

_ijerph, 2022, doi:10.3390/ijerph19073836_

Round 1

Reviewer 1 Report

You state that: "The study is part of the project “EMMA”." Please, quote the web-page of this project. The resuts should be reported only in past simple tense. So the sentence: "As all variables are non-normally distributed, a Mann–Whitney U test, respectively a Kruskal-Wallis H test, was applied." should be As all variables were non-normally distributed, a Mann–Whitney U test, respectively a Kruskal-Wallis H test, was applied. In the text, you state that "MAXQDA 2020 (VERBI Software, 2019) was used 270 for qualitative data analysis." It should be included in References as VERBI Software. (2019). MAXQDA 2020 [computer software]. Berlin, Germany: VERBI Software. Available from maxqda.com. as suggested by the web-site of this software https://www.maxqda.com/blogpost/how-to-cite-maxqda In the abstract, you state that "In all, 115 home health care workers (mean = 47.83±11.72; 81.75% female) filled in 15 the questionnaires and 237 diary entries were made by 23 persons (mean = 46.70±10.40; 91.30% female)." It is not clear what "mean = 47.83±11.72; " signifies. Later, from Table 1, it becomes clear that this is mean age. Please, revise the sentence in the abstract to In all, 115 home health care workers (mean age = 47.83±11.72; 81.75% female) filled in 15 the questionnaires and 237 diary entries were made by 23 persons (mean age = 46.70±10.40; 91.30% female). The sentence "Within 33 out of 237 diary entries, participants report the absence of joyful moments." should be Within 33 out of 237 diary entries, participants reported the absence of joyful moments. The sentence "Home health care worker identify 60 289 different types of joyful moments." should be Home health care workers identified 60 289 different types of joyful moments. The sentence "In sum, these joyful moments are mentioned 334 times." shoud be In sum, these joyful moments were mentioned 334 times. The sentence "Within these categories, joyful moments related to patients are mentioned most often, followed by joyful moments referring to the team and relatives of the persons in need of care." should be Within these categories, joyful moments related to patients were mentioned most often, followed by joyful moments referring to the team and relatives of the persons in need of care. The sentence "Joyful moments related to a sense of achievement is mentioned most often." should be Joyful moments related to a sense of achievement were mentioned most often. The sentence "To have an easy working day without any complications and valuing the own work are mentioned twice respectively third often." should be To have an easy working day without any complications and valuing the own work were mentioned twice respectively third often. The sentence "Further, experience varied, interesting and new work contents are reported as joyful" should be Further, experience varied, interesting and new work contents were reported as joyful. The sentence "Aspects related to an easy schedule, such as no time pressure, earlier or timely end of work or later beginning of the working day are reported most often." should be Aspects related to an easy schedule, such as no time pressure, earlier or timely end of work or later beginning of the working day were reported most often. The sentence "Within these categories, joyful moments related to a quiet traffic situation are mentioned most often, followed by good weather and a well-prepared car." should be Within these categories, joyful moments related to a quiet traffic situation were mentioned most often, followed by good weather and a well-prepared car. The phrase "...(e.g. in 75.5% of diary entries produced by woman, joyful moments related to the category relationships are mentioned)." should be ...(e.g. in 75.5% of diary entries produced by woman, joyful moments related to the category relationships were mentioned). In Table 3, when p = .000 it should be reported as p < .001, accordingto APA style. The sentences "Significant differences related to gender are obtained for resilience, meaning of work, and work content, each with higher values for female participants. For age, significant differences are found in work content, with decreasing percentages along increasing age. Participants working in rural regions show significantly higher values in meaning of work and mention joyful moments related to relationships significantly more often. Whereas, work content is mentioned significantly more often in diary entries recorded by employees in urban regions. Between health care workers and administrative staff, significant differences are obtained for the mention of joyful moments related to relationships, with higher values for health care workers, and related to self-care, with higher percentages of self-care related diary entries in administrative staff. No differences can be found for native language." should be Significant differences related to gender were obtained for resilience, meaning of work, and work content, each with higher values for female participants. For age, significant differences were found in work content, with decreasing percentages along increasing age. Participants working in rural regions showed significantly higher values in meaning of work and mentioned joyful moments related to relationships significantly more often. Whereas, work content was mentioned significantly more often in diary entries recorded by employees in urban regions. Between health care workers and administrative staff, significant differences were obtained for the mention of joyful moments related to relationships, with higher values for health care workers, and related to self-care, with higher percentages of self-care related diary entries in administrative staff. No differences were found for native language. If the effect sizes are also reported in the above sentences, the scientific contribution of this article would be higher. Table 3 is well-structured, but typically in scientific articles, not only p-values are reported, but also the statistical coefficients (t, F or chi in this case), together with their degrees of freedom. This permits the data to be checked. Please, report the results in past simple tense, as it should be in scientific articles. The sentences "The mean scores for resilience and meaning of work are significantly higher in female home health care workers, than in male. In addition, participants working in rural areas show significant higher levels in meaning of work, than their urban colleagues do." should be The mean scores for resilience and meaning of work were significantly higher in female home health care workers, than in male. In addition, participants working in rural areas showed significant higher levels in meaning of work, than their urban colleagues did. Please, revise the next sentences in Discussion part following this model of reporting the results only in past simple tense, because it is known when the study was conducted.

Reviewer 2 Report

The manuscript presents an interesting study conducted with a mixed quantitative and qualitative methodology on the subjective (positive) experience of nurses during the current pandemic.
I rate the manuscript as having a good level of quality, both in terms of methodology and in terms of presentation of results and discussion.
Therefore, I would suggest accepting it in the present form.

Author Response

We would like to thank you for your willingness and your time to read the manuscript. Thank you so much for your appreciative feedback on the quality of our study and for valuing our field of work. We hope that we can make with this article a valuable contribution to the scientific nursing literature and maybe encourage health care workers to recognize more often the other side of the coin to see all the joyful moments every day.

Reviewer 3 Report

Review of: The second side of the coin – resilience, meaningfulness and joyful moments in home health care workers during the COVID-19 pandemic

With the toll that the ongoing pandemic has taken on healthcare workers, this topic of resilience, meaningfulness and joy in home health nurses is important. There were no ethical concerns raised.

There has been increased interest in salutogenesis, and health/promotion in the last decade or so. While Cottrell’s concept analysis on joy and happiness lays out  the dearth of literature and works on the positive attributes of being a healthcare worker, in the 6 years since it was written, things have changed significantly. Rather than, “In the last decades, nursing literature has predominately focused on describing the prevalence of work strains and their consequences rather 30 than exploring work related resources”, it could be stated while that was the case for several decades, things have been changing in the last decade or so, however complicated by the pandemic.

While focusing on the positive, and the joyful moments, it is possible that we are challenging the narrative. We are shifting from perceiving work to be stressful, pausing long enough to take the time to reflect on the simple things that bring us joy, and therein creating meaning and sense-making. It is a conscious shift that takes effort at least initially, and then can gather momentum.  However, one can also argue that when one is feeling overwhelmed and anxious due to the uncertainty of the pandemic, does it take enormous strength to find joy? How does one replenish those reserves of resilience?

The response rate is stated to be 57.5%, with 115 home health care workers responding to the questionnaire. However, only 23 participants produced 237 diary entries – does that mean others did not? Was it because they were too stressed? What were the daily routines of those who found joy? The comments were interesting to read, and it would have been more powerful to add how some found joy amidst the pandemic.  
